# Vacuum ultraviolet spectroscopy of the lowest-lying electronic state in subcritical and supercritical water

Timothy W. Marin[1], Ireneusz Janik[2], David M. Bartels[2] & Daniel M. Chipman[2]

The nature and extent of hydrogen bonding in water has been scrutinized for decades, including how it manifests in optical properties. Here we report vacuum ultraviolet absorption spectra for the lowest-lying electronic state of subcritical and supercritical water. For subcritical water, the spectrum redshifts considerably with increasing temperature, demonstrating the gradual breakdown of the hydrogen-bond network. Tuning the density at 381 °C gives insight into the extent of hydrogen bonding in supercritical water. The known gas-phase spectrum, including its vibronic structure, is duplicated in the low-density limit. With increasing density, the spectrum blueshifts and the vibronic structure is quenched as the water monomer becomes electronically perturbed. Fits to the supercritical water spectra demonstrate consistency with dimer/trimer fractions calculated from the water virial equation of state and equilibrium constants. Using the known water dimer interaction potential, we estimate the critical distance between molecules (ca. 4.5 Å) needed to explain the vibronic structure quenching.

[1] Department of Chemistry, Benedictine University, 5700 College Road, Lisle, Illinois 60532, USA. [2] Notre Dame Radiation Laboratory, Notre Dame, Indiana 46556, USA. Correspondence and requests for materials should be addressed to T.W.M. (email: tmarin@ben.edu).

It is generally accepted that hydrogen bonding (H bonding) gives rise to the many interesting and oftentimes anomalous properties of water. However, debates continue over an exact definition of H bonding in water. This is particularly true of high-temperature (subcritical) and supercritical water (SCW), which have increasingly been touted as alternative reaction media for green chemistry[1,2], and whose participation in fundamental geochemistry[3–5] and abiogenesis[6,7] are considered crucial. The ties between H bonding and the optical properties of water have been an item of discussion over many decades, including their effect on the vacuum ultraviolet (VUV) absorption bands[8–16]. Considering that the energies and dynamics associated with excitation of these bands are responsible for the production of OH radicals, their quantitative analysis may have implications in radiation chemistry, atmospheric chemistry, radiation biology and astrochemistry.

The optical properties and electronic absorption spectrum of liquid water have previously been studied over a wide range of photon energies, from about 4–150 eV. Bernas, *et al.*[8] have reviewed much of this literature. The broad lowest-energy 'first continuum' peak in the spectrum occurs at 7.45 eV in the gas phase (Fig. 1)[17], and shows vibronic structure (Supplementary Note 1). The absorption has been attributed to an $n$ to $\sigma^\star$ ($\tilde{A}^1B_1 \leftarrow \tilde{X}^1A_1$) excitation, involving non-bonded electrons localized on the oxygen atom. The $^1B_1$ surface is also mixed with a 3s Rydberg state, making the $\tilde{A}$ excited state quite diffuse. For the liquid phase, this peak is notably broadened and shifted to the blue, with an absorption maximum of 8.3 eV (ref. 18). The origin of the observed blueshift has been assigned to H-bond solvation[19–21], Rydbergization[22–24] and/or excitonic effects[25–27]. Multiple electronic structure studies have been performed via X-ray spectroscopy that ultimately address the same excited electronic states. Much of the pertinent literature has been summarized in reviews by Nilsson, *et al.*[28], and Fransson, *et al.*[29]

Previous examinations of the temperature dependence of the liquid water absorption band up to 80 °C indicated a redshift of the absorption onset with increasing temperature[30–32]. This was correlated with a reduction in the extent of H bonding due to increasing orientational disorder. It remained a question whether this temperature shift would continue at higher temperatures and

whether changes would occur in the absorption upon reaching the water critical point ($T_c = 374$ °C, $p_c = 220$ bar). Since SCW is highly compressible, a question also lies in whether tuning the water density can induce changes between liquid-like and gas-like behaviour, that is, causing the spectrum to redshift with decreasing density as it becomes more gas-like. Measurements of the ultraviolet absorption edge were extended up to 400 °C, predominantly along a 250-bar isobar and over the wavelength range 1,900–3,500 Å, illustrating that the spectral redshift continues with increasing temperature[15]. Arguments were presented supporting the conclusion that this is a true shift in the water electronic transition energy; it cannot be explained by growth of vibrational hot bands or as a scattering artefact. The density dependence of the VUV spectrum near and above the critical temperature was also explored, and a slight blueshift of the absorption band was noted with decreasing density, likely due to a narrowing of the spectrum caused by changes in the local solvent environment. However, these conclusions were drawn based on examination of the absorption edge, and the position of the absorption maximum was merely extrapolated based on fitting. To confirm whether the entire absorption band is truly shifting, the entire spectrum has still required measurement so that the peak energy of the spectrum can be observed. In complementary X-ray absorption studies examining the same water electronic states, the pre-edge absorption ($4a_1 \leftarrow 1s$) of supercooled liquid water has been observed to similarly redshift with increasing temperature[28]. This has been interpreted as clear weakening of the H-bond network at higher temperatures[33].

Direct measurement of the condensed-phase water VUV absorption spectrum is very difficult in practice, and consequently there has been a lack of experimental study (Supplementary Note 2). Experimental investigations of SCW remain even scarcer due to the inconveniently high pressures and temperatures. In this manuscript, we present a VUV absorption experiment that has investigated the water first continuum transition from room temperature up to and exceeding the critical temperature, and for SCW as a function of pressure/density. For subcritical water, the absorbance spectrum gradually redshifts as a function of increasing temperature, consistent with the purported weakening of the H-bond network. Combining these subcritical data with the near-UV data of Kröckel and Schmidt[34], we provide extinction coefficients from 1,440–3,500 Å over the entire temperature range examined. Above the critical point, the gas-phase water monomer spectrum is replicated in the low-density limit, where H bonding is minimal. With increasing density, the spectrum gradually blueshifts, presumably due to a combination of two things: (1) an increase in the extent of H bonding, which lowers the ground-state energy, and (2) increasing Pauli repulsion[35], excitonic effects[25,26], and Rydbergization[22–24] in the more diffuse excited state, which raise the excited-state energy. Furthermore, with increasing density we find a gradual quenching of the vibrational structure inherent to the gas-phase water spectrum (Supplementary Note 1), which we analyse in terms of perturbation of the monomer excited state that could also contribute to the spectral blueshift.

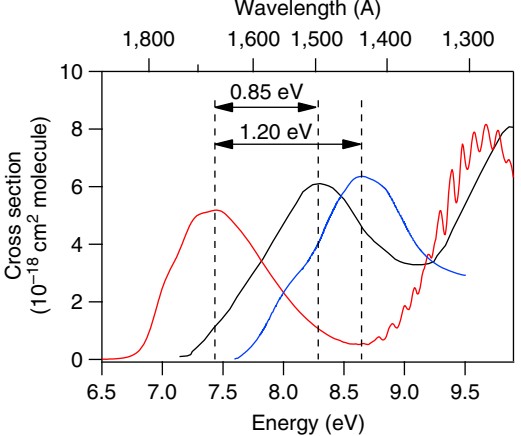

**Figure 1 | Water VUV absorption spectra from the literature.** Comparison of absorption spectra from the literature showing the lowest-lying electronic transition in the vacuum ultraviolet for gas-phase (red line)[17] and liquid phase (black line)[18] water, and for water ice (blue line)[36]. Note the considerable blueshift in the condensed-phase spectra, attributed to extensive H-bond coordination lowering the ground-state energy, and introducing solvation, Rydbergization, Pauli repulsion and excitonic effects on the excited state.

## Results

**Subcritical water**. A summary of the VUV spectra acquired for subcritical water are illustrated in Fig. 2. Note that the relative absorbance of each spectrum has been normalized to account for the differences in density (concentration) when calculating the molar extinction coefficient, $\varepsilon$. The room-temperature spectrum acquired at 23 °C shows a peak at 8.46 eV with $\varepsilon = 1{,}590\,\text{M}^{-1}\,\text{cm}^{-1}$. We note that the extinction coefficient confirms values from previous work[18], however the peak energy is observed to be

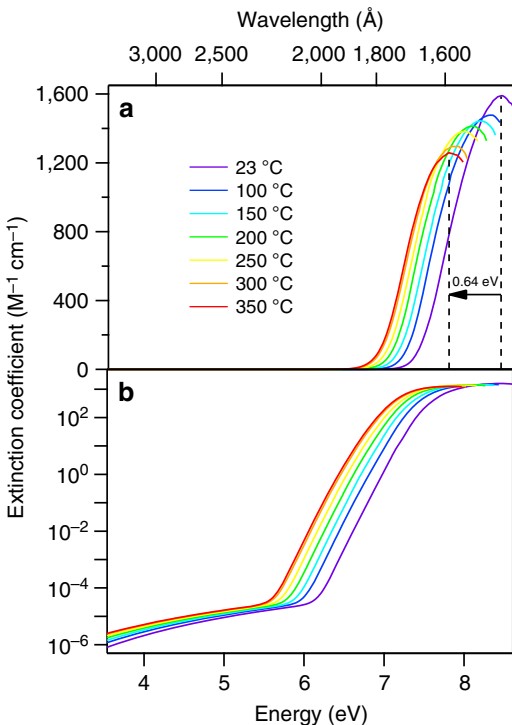

**Figure 2 | Temperature-dependent VUV absorption spectra for subcritical water.** Absorption spectra obtained for subcritical water demonstrate a clear redshift with increasing temperature. The illustrated spectra have been combined with the energy-shifted near-UV data of Kröckel and Schmidt[34]. (**a,b**) Show identical data, except on linear and logarithmic scales, respectively, to illustrate the various intensity regions of the data.

considerably higher than that obtained in reflectance spectroscopy, where Ikehata, *et al.*[14], and Kerr, *et al.*[18], reported values of 8.37 and 8.30 eV, respectively. We attribute these differences to the necessary Kramers–Kronig transformations and analysis inherent to these other studies, whereas our transmission measurement is direct.

The entire spectrum redshifts linearly with increasing temperature, in agreement with previous predictions[14,15,30–32], and by 350 °C it has shifted by 0.64 eV to lower energy compared to room temperature. The energy of maximum absorbance, $E_{max}$, steadily decreases by $-0.00203$ eV °C$^{-1}$ (16.4 cm$^{-1}$ °C$^{-1}$) (Supplementary Fig. 1), also in reasonable agreement with previous experimental approximations[14,15,34] and theoretical predictions[21]. The maximum extinction coefficient steadily decreases linearly ($\varepsilon_{max} = -0.965T + 1{,}598$) with increasing temperature, and by 350 °C reaches ~80% of its room-temperature value (Supplementary Fig. 2a).

The low-energy side of the spectra aligns very well with our previous studies of the band edge[15], and the slope of the rising edge shows excellent agreement with recent work by Kröckel and Schmidt[34]. At every temperature, the extinction coefficient of our data combined with energy-shifted Kröckel and Schmidt data (Fig. 2) can be fit surprisingly well by a sum of four-Gaussian functions, that is,

$$\varepsilon(E) = \sum_i A_i \exp\left[-\left(\frac{E - E_{max,i}}{\sigma_i}\right)^2\right], \qquad (1)$$

with $i = 0...3$. This function fits the experimental data extremely well over the entire wavelength range covered by both data sets, with deviations within 0.10 M$^{-1}$ cm$^{-1}$. A sample spectrum and

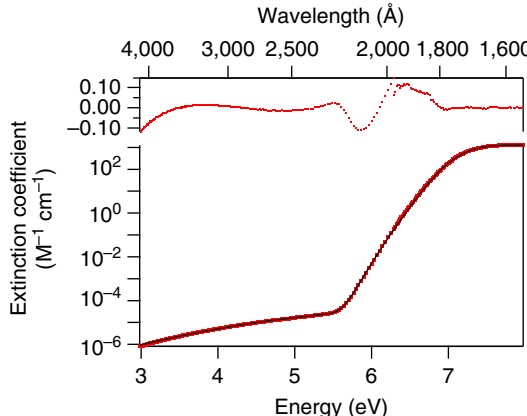

**Figure 3 | Comparison of water VUV data with near-UV data.** Here the water VUV absorption spectrum at 350 °C (red points) has been combined with energy-shifted near-UV data of Kröckel and Schmidt[34], and four-Gaussian fit (black line). Note the logarithmic scale used to facilitate visualization of the low-extinction rising edge of the absorption band. The residual of the fit is shown at the top, showing reliable reproduction of the experimental data within ±10%.

fit are shown in Fig. 3, and fitted parameters for $A$, $E_{max}$, and $\sigma$ can be found in Supplementary Table 1, along with the lowest wavelength for which the fit is reliable. These parameters can be used to generate extinction coefficients for the water UV spectrum from room temperature to 350 °C, over the wavelength range 1,440–3,500 Å. Applying fits to all the data, the overall spectral shape is well-preserved, perhaps gradually narrowing by only 5% at 350 °C compared to room temperature, confirming previous claims that the Franck–Condon envelope changes little with temperature, and suggesting there is little change to the nature of the electronic transition. Further details regarding the integrated absorption and oscillator strength are provided in Supplementary Note 3 and Supplementary Fig. 2b.

It is likely that the redshift with increasing temperature correlates with weakening of the water H-bond network due to added thermal energy (Supplementary Note 4). We make the following argument with a traditional tetrahedral coordination picture for water in mind. Hexagonal water ice, which has tetragonal H-bond coordination and shows an absorption maximum at 8.65 eV (ref. 36), can be considered a limiting case where the maximum of four H bonds per molecule is achieved. Considering that the gas-phase absorption maximum lies at 7.45 eV (ref. 17) (corresponding to zero H bonds), the 1.20-eV difference in these two peak values is a direct consequence of the H-bond influence on both the ground- and excited-state energies, and thus can serve as a rough calibration of the number of H bonds per molecule. We note that this value is in qualitative agreement with the difference in excitation energies for water pentamer (8.32 eV) and monomer (7.53 eV) obtained via EOM-CCSD/CBS calculations[37]. Dividing the difference in the peak energies by four gives an average energy shift of 0.30 eV for each H bond. We must clarify that the calculations just mentioned clearly demonstrate that most of the observed blueshift arises from accepting two H bonds, with relatively minor changes in spectral energy from the two donor H bonds. So deducing an average of 0.30 eV, while true, is a gross oversimplification of what in reality is an unknown bimodal distribution of H bond types and energies which becomes increasingly important to consider when single water moieties take on just one or two H bonds, that is, at the highest temperatures. Nonetheless, using this simple metric, the 23 °C spectrum reported in this work with an absorption maximum at

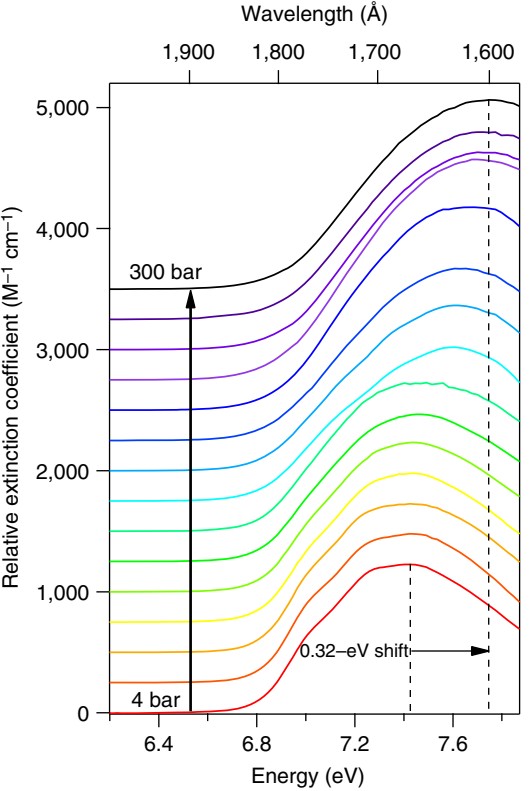

**Figure 4 | VUV absorption spectra of supercritical water at 381 °C.**
SCW spectra obtained at 381 °C gradually redshift with increasing density/pressure. The red spectrum at the bottom of the figure was acquired at 4 bar, and pressure increases for each spectrum from bottom to top in the figure as listed in Supplementary Table 2, the black spectrum being acquired at 300 bar.

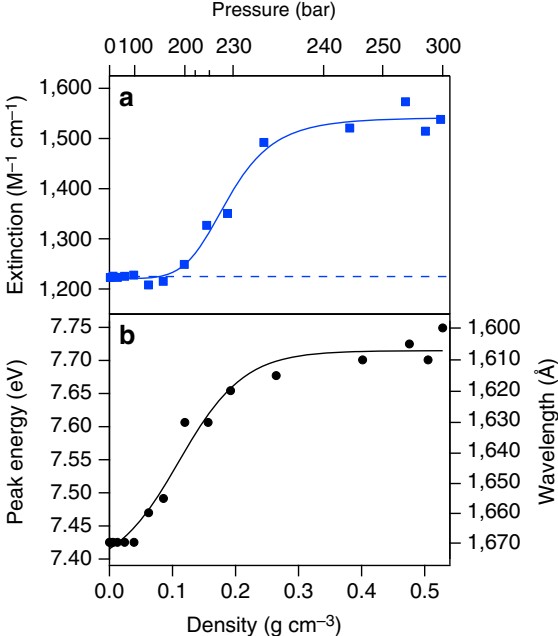

**Figure 5 | Supercritical water extinction coefficient and energy of maximum absorption.** The extinction coefficient (**a**) and energy of maximum absorption (**b**) are shown for SCW at 381 °C as a function of density and pressure. The dashed blue line indicates the expected high-density water extinction coefficient based on extrapolation of the subcritical data. The solid lines are overlaid merely to guide the eye.

8.46 eV corresponds to an average of 3.4 H bonds per molecule, and the 0.64-eV redshift that we observe in warming subcritical liquid water from 23 to 350 °C corresponds to breaking approximately 2.1 H bonds overall, with an associated loss of 0.0065 H bonds per °C. Thus, at 350 °C, an average of 1.2 H bonds per molecule remain. We note that these values agree quite well with those reported in a variety of other experimental studies and simulations; see reviews by Kalinichev[38], Chaplin[39] and Rastogi, et al.[40], and references therein.

Previous matrix isolation spectroscopy studies of water in Ne, Ar and Xe rare-gas matrices demonstrated large spectral blueshifts with decreasing distance between individual matrix atoms, that is, the matrix cavity size occupied by individual water molecules[22,24]. This is interpreted in terms of restricting the volume of the 3s Rydberg-character excited wavefunction, and the magnitude of the shift in Ne is surprisingly almost identical to liquid water. Nevertheless, we are convinced that the Rydbergization argument does not apply to liquid or solid water. For liquid water, it is clear that the optical transition is blueshifted due to unfavourable solvation of the excited state and stabilization of the ground-state orbitals by H bonds. Each water molecule is surrounded by others having nearly degenerate empty orbitals with which the excited Rydberg 3s electron can interact. In the rare-gas matrix, these unoccupied orbitals simply do not exist.

**Comparison of SCW to subcritical and gas-phase data.** SCW absorption spectra were acquired at 381 °C over 15 different pressures ranging from 4 to 300 bar, with corresponding densities ranging from 0.0013 to 0.529 g cm$^{-3}$, as listed in Supplementary Table 2 and illustrated in Fig. 4. At the lowest pressures, the SCW data should mimic gas-phase data, where there is very minimal interaction between water molecules due to the low density. At 4 bar the overall shape of the room-temperature gas-phase spectrum[17] is largely reproduced (Supplementary Fig. 3). The symmetric stretch vibrational progression[41–49] is obvious, though slightly less pronounced than in the room-temperature gas-phase spectrum, and the spectral width (FWHM) of 0.93 eV is matched. Conversely, the peak of the low-density SCW spectrum is red-shifted by 0.017 eV (140 cm$^{-1}$) compared to the room-temperature gas-phase spectrum. After re-examining the calibration of the monochromators used to collect these data and communicating with the original authors of the gas-phase spectrum regarding similar calibrations, we are convinced that all the data are correct, and the peak energy discrepancy is physically justifiable (Supplementary Note 5). One other striking difference between the low-pressure SCW data and the gas-phase data is the 10% lower extinction coefficient for SCW. Certainly, a drift of ± 0.2 bar in our pressure control would be reasonable over the course of our measurements for a particular spectrum, which would account for at least 5% of the discrepancy. It is entirely possible that the pressure transducer used for these measurements, which is rated as linear from 0–345 bar, is not optimal for use at pressures as low as 4 bar, as only ~1% of its dynamic range is being utilized. Therefore we suggest that our data basically agree with those of Mota, et al.[17] Certainly more confidence should be placed in the accuracy of the latter measurements, which were designed specifically to obtain excellent results for very low-density gas-phase water, whereas our experiments by design address a very broad range of densities, with better pressure control capabilities at higher pressures.

The peak SCW extinction coefficient is shown as a function of density in Fig. 5a. It was already mentioned that on the

low-density end, where the data should predominantly show gas-phase behaviour, the measured extinction coefficient of $1,225 \, M^{-1} cm^{-1}$ undershoots the gas-phase literature value[17] by 10%. Considering the linear decrease in $\varepsilon$ with increasing $T$ for subcritical water (Supplementary Fig. 2a), one can extrapolate the condensed-phase value to be $1,230 \, M^{-1} cm^{-1}$ at 381 °C. Our SCW data disagree with this prediction, with a measured value of $\sim 1,550 \, M^{-1} cm^{-1}$ at the highest densities that is quite close to that of the room-temperature liquid. The only way we can explain this 20% difference is to consider the markedly different local environment for SCW (mix of monomers and small clusters with large empty spaces between them) compared to pure gas-phase or condensed-phase water. It might be reasonable to expect that the VUV refractive index is also markedly different for SCW. Based on the trends that can be extracted from simple field models[50], it could be that the VUV refractive index increases significantly with increasing pressure, compensating for the observed increased extinction coefficient at high densities. Since, to our knowledge, these refractive index data do not exist in this wavelength domain for SCW, we unfortunately cannot address this discrepancy further.

With increasing pressure/density, the SCW spectrum broadens and blueshifts by 0.32 eV at 300 bar compared to the 4-bar spectrum, as seen in Fig. 4 and the Fig. 5b. We ascribe this to additional contributions from 'condensed-phase' behaviour, that is, dimers, trimers, etc., that lower the ground-state energy, working in concert with enhanced Rydbergization[22–24], excitonic[25,26] and Pauli repulsion[51] effects in the more diffuse excited state, which become increasingly important at higher densities. Though no experimental electronic spectra are known to exist for isolated dimer or other small cluster species, the extent of blueshift imparted by these H-bonded species has been examined computationally[37,52–54]. An extrapolation of the trend in $E_{max}$ versus $T$ (Supplementary Fig. 1) predicts that the peak energy at 381 °C for a condensed-phase spectrum would be 7.72 eV. Our highest-density SCW data agree reasonably with this extrapolation, with a value of 7.75 eV. Turning to our previous arguments for subcritical water, the energy involved in the shift as a function of density/pressure implies forming $\sim 1.1$ H bonds per molecule in compressing from 4 to 300 bar, as delocalization of the exciton is gradually facilitated by increasingly proximate molecules. To our knowledge, the number of H bonds present in SCW over a wide density range near 381 °C has not been reported. However, as recently summarized by Marcus[55], several studies at 400 °C indicate possible formation of up to 1.4 H bonds at densities similar to those examined in our work, in general agreement with our results. It is very difficult to differentiate excitonic versus Rydbergization effects in the low-density regime. Our hope is that future studies of SCW mixed with rare-gas species might help isolate any Rydberg contribution.

Returning attention to Fig. 4, the vibronic structure inherent to the 4-bar spectrum is gradually extinguished with increasing pressure/density. It is highly likely that the dimers and trimers that form with increasing pressure modify the excited-state surface, smearing out the coherence tied to the symmetric stretch coordinate. The diminishing contribution of the vibronic structure becomes obvious in inspecting the derivatives of the SCW spectra, as shown in Fig. 6. The structure is essentially not present for pressures above 170 bar. Taking the second derivative removes the broad background feature, leaving the oscillatory features in place, though unfortunately also increasing the amplitude of the point-to-point random noise. The relative amplitude of the oscillatory features can then be measured relative to the background noise in the data by summing the deviation from zero across the spectrum. Doing so demonstrates an exponential decay in these deviations, and hence the extent of

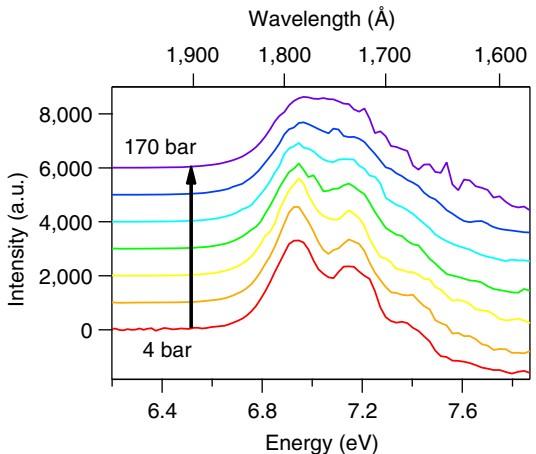

**Figure 6 | Quenching of vibronic structure in supercritical water.** First-derivative SCW absorption spectra are shown for pressures from 4 bar (red curve) to 170 bar (purple curve), which better illustrate the vibronic structure inherent in the spectra. It is obvious that the structure diminishes in intensity with increasing pressure/density, and by 170 bar its contribution is essentially gone.

vibronic structure, with increasing density. Defining the contribution of the vibronic structure as unity in the 4-bar spectrum, the amplitude falls off as $e^{-23.15\rho}$, as shown in Fig. 7a, where $\rho$ is the SCW density in g per $cm^3$ as listed in Supplementary Table 2.

**SCW dimer and trimer equilibrium constants**. Considering the possible collective contribution of various water moieties in producing the experimental SCW spectra, we turn to known equilibrium properties to investigate whether their individual contributions can be extracted from the data. We have tabulated the relative equilibrium populations of SCW monomers, dimers and trimers as a function of density for each of our experimental spectra based on recent work by Tretyakov, et al.[56–58] In this work the authors formulated virial coefficients and equilibrium constants for water dimerization and trimerization based on the most comprehensive compendium of water thermochemical properties to date[59] as well as infrared work by Vigasin, et al.[60] The populations of total monomer, dimer and trimer are given in Supplementary Table 2, and illustrated as a function of density and pressure in Fig. 7b. We have no reason to doubt the credibility of these data, so we take the numbers at face value. They provide a much-needed anchor for the analysis described below.

One should note an immediate discrepancy between the monomer/dimer/trimer fractions shown in Supplementary Table 2 and Fig. 7, compared to the derivative analysis based on Fig. 6. The derivative analysis implies that the vibronic structure due to the monomer 'gas-phase' contribution to the spectrum is damped out at pressures above 170 bar, whereas the tabulated monomer/dimer/trimer fractions indicate that the monomer should be the dominant species under all experimental conditions. Even at our highest pressure of 300 bar, we should expect 60% of the monomer symmetric stretch vibrational structure to persist. It is simply not present.

Clearly there is disagreement in the literature regarding the extent and nature of H bonding in SCW (Supplementary Note 6). Considering the wide variety of results and the added complexity of our new direct measurements of the SCW electronic spectrum, this calls the question exactly what constitutes a water monomer versus a dimer? In addition to the H-bonded global minimum

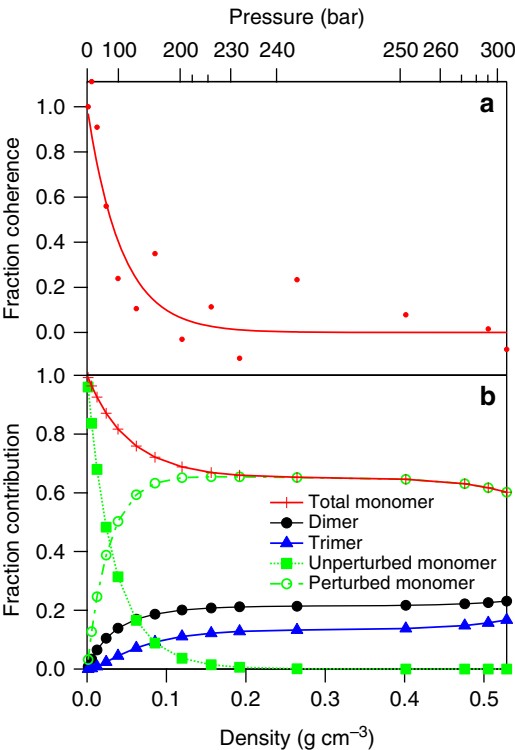

**Figure 7 | Fraction of observed coherence at each supercritical water density.** Compared with relative fractions of monomer, dimer and trimers. (**a**) Shows an exponential fit to the fraction of observed vibrational coherence in the SCW spectra as a function of density. (**b**) Displays the relative fraction populations of SCW monomers (solid red curve), dimers (black curve) and trimers (blue curve) for SCW at 381 °C as a function of density/pressure based on the information provided by Tretyakov, *et al.*[56–58] Calculated fractions of electronically unperturbed (dotted green line) and perturbed (dash-dot green curve) monomer are shown as well, based on analysis of **a**, as described in the text. Experimental pressure/density points are represented with symbols on each curve.

dimer structure, many other configurations of two water molecules will be energetically attractive and so have significant, albeit transient, existence, thereby acting as effective dimers[52].

We are thoroughly confident that the water thermochemical data are correct, so the observed disappearance of the monomer symmetric stretch vibrational signature with increasing density/pressure in SCW apparently does not correspond to the formation of textbook dimers and trimers. At the highest pressures measured, there should be a large amount of monomer where we observe none. Additionally, the thermochemical data should be able to predict the average number of H bonds per molecule if a weighted average is taken over the fractions of monomer (0 H bonds), dimer (1 H bond) and trimer (2 H bonds) species. At 300 bar, the fractions listed in Supplementary Table 2 dictate an average of 0.36 H bonds per molecule.

Infrared data acquired at 377 and 400 °C indicate breakdown of rotational structure at densities as low as $0.01 \, \text{g cm}^{-3}$ (refs 56,60), where monomers should still make up 94% of the mix and there is very minimal H bonding. While the presence of H-bonded true dimer species would clearly have strong impact on the rotational contributions to the spectra, not enough of them are present to justify the extent of structural loss. Under such conditions and still higher densities, a dimer might be envisioned which is not necessarily H-bonded, but is dipole-bound. Tretyakov, *et al.*, discuss this idea at some length[56–58]. Many

highly excited quasibound or metastable states of the dimer might exist with energies higher than the ground-state dimer dissociation energy, and where the individual monomers of the dimer can rotate almost freely. Considering that the long-range interaction potential is definitively electrostatic in nature, the two monomer molecules will perturb each other's excited electronic states at non-equilibrium geometries and distances as well. For such states, the individual water molecules could have almost the same structure and electronic absorption spectrum, though the spectrum would be additionally blueshifted and broadened by the state lifetime. Moreover, considering the myriad possible geometrical arrangements of such states with broad local environmental distributions, one might expect the associated spectral signatures to illustrate significant inhomogeneous broadening.

If we consider the vibrational coherence feature to be characteristic of electronically 'unperturbed' monomers, we can then subtract this unperturbed monomer fraction from the total fraction of monomer tabulated in Supplementary Table 2 to give a fraction of 'perturbed monomer' as a function of pressure/density. These values are also shown in Supplementary Table 2 and are plotted in Fig. 7b. In essence, we assert that all water monomer species present at densities exceeding ca. $0.1 \, \text{g cm}^{-3}$ have substantial perturbation of their excited-state surfaces compared to low-pressure SCW.

**Non-H-bonded SCW interactions and coherence quenching.** Electronic structure calculations were carried out to estimate the effect of transient water–water interactions on the spectrum. Most transient interactions will be of dipole–dipole nature between two monomers. The maximum dipole–dipole interaction is obtained by placing two monomers in *pi*-stacked parallel planes with oppositely-oriented dipoles. At the optimal geometry so-constrained, it was found that the potential is attractive with binding energy of 0.06 eV, which is about one-fourth of the global minimum potential with H bonding. The lowest vertical excitation energies associated with the two monomers were 0.05 eV below and 0.24 eV above that of the isolated monomer, their average then being blueshifted by 0.10 eV from that of the isolated monomer. One of the two associated oscillator strengths was exactly zero by symmetry, while the other was slightly more than double that of the isolated monomer, their average being just 6% above that of the isolated monomer. A significant range of other non-optimal *pi*-stacked geometries showed effects roughly half as large as at the optimum *pi*-stacked geometry. It may be concluded that attractive dipole–dipole interactions between monomers could blueshift their average excitation energy by as much as 0.1 eV and shift their combined oscillator strength by as much as 6%. However, most such transient interactions will probably cause substantially smaller shifts.

In order for the symmetric stretch progression to be observed in the $\tilde{A}^1B_1 \leftarrow \tilde{X}^1A_1$ absorption spectrum, wavepacket recurrence is necessary. The presence of a second water molecule nearby will certainly perturb symmetry and electronic structure, but how sensitive is the recurrence to this perturbation? How close does the second molecule need to be? We can estimate a critical radius, $R_c$, for quenching the symmetric stretch from the exponential behaviour illustrated in Fig. 7a using a combination of Poisson statistics and the high-quality water–water dimer potential of Jankowski and coworkers[61] (Supplementary Note 7). This provides the running coordination number from the dimer potential (Supplementary Fig. 4). In short, we find that $R_c \sim 4.5$ Å, significantly larger than the experimentally known equilibrated dimer separation of 2.98 Å (ref 62). A reasonable interpretation is that the large excited-state orbital must be perturbed sufficiently

at this long interaction distance to quench any vibrational recurrence at the saddle point.

**SCW spectra model.** All we have discussed thus far demands fitting the experimental SCW spectra to a model that reflects these ideas. We therefore fit the total SCW spectrum at every pressure/density with a function that is essentially the sum of the unperturbed monomer, perturbed monomer, dimer and trimer spectral contributions, weighted by their individual fractions. For the unperturbed monomer spectrum, we applied a fitted sum of five Gaussians reminiscent of the condensed-phase subcritical spectra discussed earlier using equation 1, but now with $i = 0 \ldots 4$, per the values in Supplementary Table 3. The exact same unperturbed monomer spectrum was used to produce the perturbed monomer, dimer and trimer contributions, but with some small adjustments. Since the perturbed monomer should be blueshifted with respect to the unperturbed spectrum, we introduced a fitted blueshift in energy for the perturbed monomer contribution. We also introduced a Gaussian convolution to smear out the vibrational coherence feature, where the s.d. of the convolving function is a fit parameter at every pressure/density, that increases in magnitude at higher pressures/densities as the monomer spectrum becomes increasingly perturbed. The dimer and trimer spectra are identical to the perturbed monomer spectrum, except the extent of blueshifts are fixed at 0.251 and $0.549 \pm 0.1 \, \text{eV}$ respectively, with corresponding oscillator strengths 13 and 15% greater than that of the water monomer, based on our LRC-$\mu$-BOP/6–311(2 + )G($d,p$) electronic structure calculations. These are also convolved with a Gaussian to smear out the coherence feature, except the s.d. of the convolving function is fit as a shared parameter for the equilibrated dimer and trimer at every pressure/density.

The data were fit globally using a least-squares Levenberg–Marquardt fitting algorithm. The fractions of unperturbed monomer, perturbed monomer, dimer and trimer are fixed at every density using the values in Supplementary Table 2. As already mentioned, the energies of the dimer and trimer species are fixed. In all, only four parameters are fit: (1) the perturbed monomer energy shift at each pressure/density; (2) the s.d. of the necessary broadening function for the perturbed monomer at each pressure/density; (3) the s.d. of the necessary broadening function for the dimer and trimer (independent of density); (4) the overall intensity of the fitting function.

Despite the severe assumptions, the model achieves a very reasonable global fit, with best fitted dimer and trimer blueshift values of 0.301 and 0.659 eV, respectively, that is, 20% greater than their computed values. Selected fitted spectra at four pressures/densities are shown in Fig. 8. In comparing the 4-bar spectrum (which nearly entirely is ascribed to unperturbed monomer) to the 280-bar spectrum (which is ascribed to a mix of absorbing species), it is obvious that the unperturbed monomer cannot possibly be the major contributor to the 280-bar spectrum. Despite the fact that monomers overall constitute 62% of the species present at 280 bar, the characteristic monomer absorption and corresponding vibrational structure are unobserved on the red edge of the 280-bar spectrum, and there is insufficient intensity at energies less than 7.4 eV. The s.d. of the perturbed monomer broadening function and the perturbed monomer energy shift are both shown in Fig. 9. The extent of electronic perturbation with increasing pressure/density forces the monomer to gradually blueshift up to 0.18 eV at 300 bar, approaching but not surpassing the calculated 0.25-eV difference between the monomer and equilibrated dimer energies. The extent of broadening necessary to eliminate the vibronic structure similarly gradually increases with increasing pressure/density,

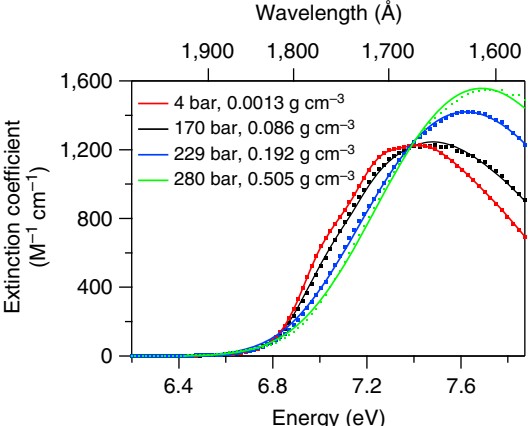

**Figure 8 | Fits to experimental supercritical water VUV absorption data at selected pressures/densities.** The spectra model described in the text nicely reproduces the water gas-phase spectrum, including its vibronic structure, in the low-density limit. It broadens and blueshifts with increasing pressure/density, as the monomer electronic structure becomes perturbed and dimers and trimers increasingly contribute, reproducing the loss of coherence with increasing density.

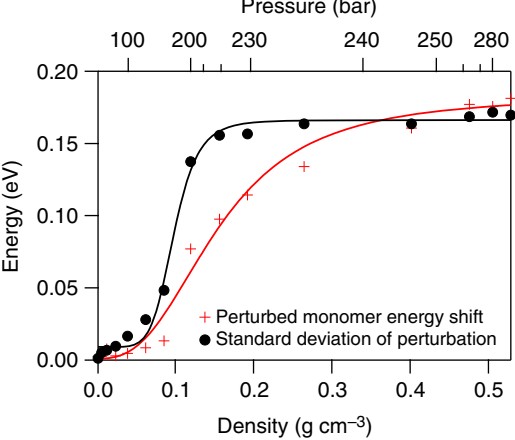

**Figure 9 | Density dependence of perturbed monomer energy shift and coherence quenching feature.** The fitted perturbed monomer energy shift is shown as a function of density, as well as the fitted s.d. of the Gaussian broadening function needed to quench the coherence feature. We note that the perturbed monomer energy shift approaches the calculated dimer shift of 0.25 eV, but never exceeds it. The curves superimposed on the fitted data are overlaid merely to guide the eye.

levelling out to a value of 0.19 eV at 300 bar. The fitted s.d. for the necessary dimer and trimer broadening is 0.21 eV.

In summary, if a significant portion of monomers having perturbed electronic structure exist, the extent of this perturbation must truly be significant. At the highest densities/pressures, a blueshift of 0.18 eV to the monomer transition energy is necessary to explain the observed energy shift of the SCW spectrum with increasing pressure/density. A substantial portion of this energy can be accounted for by the presence of dipole-bound states, which might blueshift the monomer absorption by 0.10 eV.

## Discussion
We have reported VUV absorption data on subcritical water and SCW to investigate the lowest-lying $\tilde{A}^1B_1 \leftarrow \tilde{X}^1A_1$ first continuum electronic transition. By incorporating recent

 

near-UV data[34] with our own VUV results, we have provided temperature-dependent extinction coefficients for the subcritical spectra over the entire wavelength range 1,440–3,500 Å. The 0.64-eV redshift in the absorption maximum on heating subcritical water from 23 to 350 °C correlates with breaking approximately 2.1 H bonds per molecule, which diminishes the delocalization of the water exciton. For SCW at 381 °C, the known gas-phase monomer spectrum[17] is reproduced in the low-density limit. Increasing the water density from 0.0013 to 0.529 g cm$^{-3}$ over the pressure range 4–300 bar gives rise to a spectral blueshift of 0.32 eV, correlating with formation of an average 1.1 H bonds per molecule. The vibronic structure apparent in the gas-phase spectrum due to the excited-state symmetric stretch is quenched with increasing density upon gradual formation of transient dimer species which significantly perturb the water electronic structure. Comparison of our data to a high-quality dimer potential allows us to estimate the critical distance between neighbouring water molecules required to quench the vibrational coherence. Surprisingly, this critical interaction distance is on the order of 4.5 Å, significantly larger than the equilibrated water dimer intermolecular distance. No doubt this is due to the partial Rydberg nature of the Ã excited state. We have fit the VUV SCW spectra to a model consistent with the fractions of water dimers and trimers thought to be present at each density via the water virial equation of state and dimer and trimer equilibrium constants[56,57]. For the electronic spectrum, it is clearly necessary to make the distinction between perturbed and unperturbed monomer. Rydberg and excitonic effects both must contribute to the optical spectrum, and we aspire to perform future VUV experiments that examine SCW mixed with rare-gas species in order to help separate the two.

## Methods

**VUV absorption experiment.** The nature of this high-sensitivity, high-pressure, high-temperature VUV experiment and technical details regarding the light source and detection, sample cell specifications, sample preparation, temperature/pressure control and sample flow have been published previously[63,64]. VUV measurements were carried out at the Stainless-Steel Seya beam line of the Synchrotron Radiation Center, University of Wisconsin-Madison. The combination of a sub-micron path length sample cell, synchrotron light source, and secondary filtering monochromator on the beam line allows for six orders of magnitude in light detection dynamic range. In short, spectra of a water-filled or empty sample cell were typically recorded over the range 1,400–2,200 Å in increments of 5 Å via a photon counting technique. Unfortunately, data cannot be acquired at shorter wavelengths due to the absorption onset of the sapphire widows in the sample cell. Photon counts were normalized to fluctuations in the synchrotron beam current in real time during data acquisition. Photons were generally counted for 1 s per data point. When transmittance was low ($<10^4$ counts per second), photons were counted for up to 20 s per point. Typically 30 min were necessary to acquire an entire spectrum. The beam line monochromator, secondary filtering monochromator and photon counter were controlled and synchronized through Igor Pro 6.0 (ref. 65) run on a notebook PC. The reported spectra are actually composite spectra, compiled from a variety of sample cells with variable path lengths to accommodate measuring six orders of magnitude in absorbance/extinction. Deionized water was obtained from a Millipore Milli-Q Water Purification System (18.2 MΩ cm), and sparged for at least 1 h with helium gas prior to filling the sample cell to remove any residual dissolved gases. Temperature and pressure conditions during measurements were stable within ± 0.2 °C and ± 0.2 bar, respectively. Extinction coefficients for all spectra were corrected for density changes as a function of temperature and pressure based on the water equation of state[59].

**Electronic structure calculations.** Electronic structure calculations performed to estimate the effect of transient non-H-bonded water–water interactions on the spectrum were done using the LRC-$\mu$-BOP ($\mu = 0.47$) DFT method[66,67] with the 6–31(2 +)G(d,p) basis set[68,69] in the Q-Chem 4.0 software package[70]. This combination has previously been shown[37,52] to provide excitation energies and oscillator strengths of water clusters in good agreement with much higher level methods based on EOM-CCSD with large basis sets.

**Data availability.** The data files used to prepare the figures shown in the manuscript are available from the corresponding author upon request.

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

## Acknowledgements

This work is based in part upon research conducted at the Synchrotron Radiation Center, University of Wisconsin-Madison, which was supported by the National Science Foundation (NSF) under Award DMR-0537588. We thank the SRC staff, particularly G. Rogers, M. Bissen, M. Severson, R. Julian, M. Fisher, G. Vlasak and D. Wallace. The help from machine shop staff of the Physics Department at University of Notre Dame is greatly appreciated. We thank K. Darr and M. Richmond from the Notre Dame Nanofabrication Facility for introduction to and help in metal vapour deposition techniques and instrumentation. We thank A. Ikehata for the sharing of detailed water VUV refractive index data from 10 to 70 °C and P. Limão-Vieira for his communications regarding the water room-temperature gas-phase spectrum. We are greatly indebted to P. Jankowski for providing an angular average over his recently published dimer–dimer potential, for the potential of mean force calculation. The research described herein was supported by the Division of Chemical Sciences, Geosciences and Biosciences, Basic Energy Sciences, Office of Science, United States Department of Energy, through grant DE-FC02-04ER15533. This is contribution number NDRL-5136 from the Notre Dame Radiation Laboratory. T.W.M. was funded by the Research Corporation for Science Advancement CCSA Award 7693, National Science Foundation-RUI Award 0809467 and the Benedictine University College of Science.

## Author contributions

D.M.B., I.J. and T.W.M. conceived the project. I.J. and T.W.M. designed and performed the experiments and carried out the majority of data analysis. D.M.C. performed the electronic structure calculations. All authors contributed significantly to discussion of the results and contributed to the manuscript.

## Additional information

**Competing interests:** The authors declare no competing financial interests.

