## [Peer Review File · Nature Communications]

Reviewers' comments:

Reviewer #1 (Remarks to the Author):

The authors report on and interpret VUV absorption spectra of liquid water for the lowest excited electronic state of ambient to supercritical water including, in the latter case, the dependence on density. These experimental data are valuable and should be published, but relevant complementary spectroscopic data from the x-ray region have been neglected, which, in combination with the presented data, could be used to broaden the interpretation. I will give detailed comments and suggestions below. I regard these mainly as optional and will recommend the paper for publication once they have been considered by the authors.

1) Page 2, 2nd paragraph. Important studies of the electronic structure through the absorption spectrum have been performed also in the x-ray range of photon energies, actually addressing the same excited electronic states as in the present study. Relevant references here (reviews) are Nilsson et al., *J. Electron Spec. Rel. Phen.* **177**, 99-129 (2010); Fransson et al., *Chem. Rev.* **116**, 7551-7569 (2016).

2) Page 3, 2nd paragraph. Also the pre-edge ($1s \rightarrow 4a1$) in XAS of liquid water red-shifts with increasing temperature (Nilsson et al., *J. Electron Spec. Rel. Phen.* **177**, 99-129 (2010)) which has been interpreted in terms of H-bond weakening (Nilsson and Pettersson, *Nat. Commun.* **6**, 8998 (2015)). This is the lowest excited state in the liquid both in VUV and in XAS (see also Brancato et al., *Phys. Rev. Lett.* **100**, 107401 (2008)).

3) Page 5, middle 1st paragraph. The interpretation that a blue-shift of the first excited state with increasing density should be due to enhanced H-bonding in the ground state is not necessarily true. The excited state is more diffuse than the ground state and will experience more Pauli repulsion against the environment, which will increase with increasing density. Thus, both ground and excited state effects must be considered when investigating spectral shifts. An early study of this in connection with matrix isolation spectroscopy is given by Jansson et al., *Chem. Phys.* **85**, 355 (1984).

4) Page 8, beginning 2nd paragraph. The connection between red-shift with increasing temperature and weakening of the H-bond network is made in Nilsson et al., *Nat. Commun.* **6**, 8998 (2015).

5) Page 8, 2nd paragraph. X-ray absorption spectroscopy of ambient water has, based on the same excited states (bar the difference in a core hole versus valence hole), been interpreted in terms of two structures with the dominant having only two H-bonds (Myneni et al., *J Phys: Condens. Mat.* **14**, L213-L219 (2002); Wernet et al., *Science* **304**, 995-999 (2004); Huang et al., *Proc. Natl. Acad. Sci. (USA)* **106**, 15214-15218 (2009)). The authors should discuss and comment on this alternative interpretation. It is clear that x-ray spectroscopies give a more local projection of the electronic structure due to the involvement of the core level while valence excitations integrate over a broader interaction volume. Could this explain the qualitatively different pictures of the extent of H-bonding from the two spectroscopies?

6) Page 11, lines 260-267. The discussion should take into account also the Pauli exclusion effects from comment 3 above.

7) Page 25, Figure 1. The comment on the blue-shift as due solely to extensive H-bond coordination should be modified in the light of the comments above. The lowest excited state in the liquid is into the antibonding $4a1$ state in XAS which is also the dominating character in optical spectroscopy of water (Brancato et al., *Phys. Rev. Lett.* **100**, 107401 (2008)). The blue-shift will not only relate to the number of H-bonds, but also to the strength and to the density through the excluded volume effects.

Reviewer #2 (Remarks to the Author):

Referee Report for NCOMMS-16-20440, Marin

This paper sets out some important outstanding questions on what makes water water, namely (p 1 - 2): "... However, debates continue over an exact definition of hydrogen bonding in water. ...",

and " ... Furthermore, how hydrogen bonding in a colligative sense gives rise to bulk properties is not fully understood. ..." The paper then goes on to mention sub-critical and super-critical water because of their relevance to green chemistry. Yet another thread is the connection between ultra-violet excitation of water and the production of free radicals, with "implications in radiation chemistry, atmospheric chemistry, radiation biology, and astrochemistry."

The first job of the reviewer is to ascertain whether any progress towards answering these posed questions is demonstrated in the paper.

The paper goes on to document in some detail the efforts and difficulties associated with attempting to obtain ultra-violet information about water, especially at increased temperature and pressure. Finally a new experiment is summarized using documented new methods developed at a synchrotron light source.

The results section gives an extensive account of the changes to the UV spectra with increasing temperature towards the critical point (I assume this is along the liquid-vapour equilibrium curve?) and then above the critical point at 381 C, with increasing pressure. For the former set of measurements, the spectra red-shift, with a linear decrease in extinction coefficient, suggesting a gradual reduction in the strength or degree of hydrogen bonding. For the super-critical data the spectra blue shift, and the extinction coefficient increases, in a non-linear way with increasing pressure, with some sort of transition at densities in the range 0.1 – 0.2 g/cm³. There is extensive discussion of the significant uncertainties, for example, of the extent to which water can be regarded as hydrogen bonded, associated with this region of the water phase diagram, despite many years of research. Finally attempts are made to quantify the trends seen in terms of numbers of nearest neighbour molecules as a function of density.

Yet at the end of the day, do we learn anything from the new measurements? Given all the uncertainties and mis-information about water that occurs in the literature, I would be extremely cautious about accepting on trust statements like (p 8) "It is quite possible that the extent of the red shift with increasing temperature can be correlated with weakening of the water hydrogen bond network due to added thermal energy" and (p 12) "It is highly likely that the dimers and trimers that form with increasing pressure greatly affect the electronic transition, smearing out the coherence tied to the symmetric stretch coordinate." This kind of heuristic reasoning, based on intuition rather than scientific fact, can actually be quite misleading since it is mostly not quantitative and is not supported by any kind of physical justification. Almost certainly the real system is more complicated than such simplistic statements would imply. Whereas it is relatively easy to notice and plot trends between thermodynamic variables and spectral properties, understanding what those trends actually mean in the real material is much tougher.

So I can appreciate the experimental achievement of the authors, but whether the results get us any closer to understanding the questions that are raised at the outset of the paper is not clear. The data and analysis should certainly be published, but are not appropriate for Nature Comms.

Reviewer 1

The authors report on and interpret VUV absorption spectra of liquid water for the lowest excited electronic state of ambient to supercritical water including, in the latter case, the dependence on density. These experimental data are valuable and should be published, but relevant complementary spectroscopic data from the x-ray region have been neglected, which, in combination with the presented data, could be used to broaden the interpretation. I will give detailed comments and suggestions below. I regard these mainly as optional and will recommend the paper for publication once they have been considered by the authors.

We have found the comments of Reviewer 1 to be most constructive, and have addressed them all below. We thank Reviewer 1 in that we believe his/her suggestions have strengthened the paper.

1) Page 2, 2nd paragraph. Important studies of the electronic structure through the absorption spectrum have been performed also in the x-ray range of photon energies, actually addressing the same excited electronic states as in the present study. Relevant references here (reviews) are Nilsson et al., *J. Electron Spec. Rel. Phen.* **177**, 99-129 (2010); Fransson et al., *Chem. Rev.* **116**, 7551–7569 (2016).

We admit that we were remiss in not acknowledging the significant body of work that has been performed via x-ray spectroscopy to examine water electronic structure. In light of the suggestion above, we have added appropriate text throughout the manuscript (especially see the 2nd paragraph on page 2, 1st paragraph on page 4, and 1st paragraph on page 9). We have included the suggested references.

2) Page 3, 2nd paragraph. Also the pre-edge ($1s \rightarrow 4a_1$) in XAS of liquid water red-shifts with increasing temperature (Nilsson et al., *J. Electron Spec. Rel. Phen.* **177**, 99-129 (2010)) which has been interpreted in terms of H-bond weakening (Nilsson and Pettersson, *Nat. Commun.* **6**, 8998 (2015)). This is the lowest excited state in the liquid both in VUV and in XAS (see also Brancato et al., *Phys. Rev. Lett.* **100**, 107401 (2008)).

We have now noted this pre-edge shift specifically in the 1st paragraph on page 4, and have included the suggested references.

3) Page 5, middle 1st paragraph. The interpretation that a blue-shift of the first excited state with increasing density should be due to enhanced H-bonding in the ground state is not necessarily true. The excited state is more diffuse than the ground state and will experience more Pauli repulsion against the environment, which will increase with increasing density. Thus, both ground and excited state effects must be considered when investigating spectral shifts. An early study of this in connection with matrix isolation spectroscopy is given by Jansson et al., *Chem. Phys.* **85**, 355 (1984).

What entirely agree with Reviewer 1 regarding the need to simultaneously address the ground- and excited-state contributions to the observed spectral shift, and we obviously did not convey this properly in the first draft of the manuscript. We have revised the text

accordingly (see the 1st paragraph on page 5, 2nd paragraph on page 13, and Figure 1 caption).

4) Page 8, beginning 2nd paragraph. The connection between red-shift with increasing temperature and weakening of the H-bond network is made in Nilsson et al., Nat. Commun. **6**, 8998 (2015).

This has been noted in the first paragraph on page 4.

5) Page 8, 2nd paragraph. X-ray absorption spectroscopy of ambient water has, based on the same excited states (bar the difference in a core hole versus valence hole), been interpreted in terms of two structures with the dominant having only two H-bonds (Myneni et al., J Phys: Condens. Mat. **14**, L213-L219 (2002); Wernet et al., Science **304**, 995-999 (2004); Huang et al., Proc. Natl. Acad. Sci. (USA) **106**, 15214–15218 (2009)). The authors should discuss and comment on this alternative interpretation. It is clear that x-ray spectroscopies give a more local projection of the electronic structure due to the involvement of the core level while valence excitations integrate over a broader interaction volume. Could this explain the qualitatively different pictures of the extent of H-bonding from the two spectroscopies?

We have referenced the suggested papers, and briefly discussed the two H-bond interpretation on pages 8 and 9. We apologize to Reviewer 1 in that, despite the suggestion, we have opted to avoid the temptation to overanalyze our data in light of the two H-bond interpretation, as apparent controversy has surrounded these ideas for some years. It is not our intention to settle this controversy, and as such we have interpreted our data in the context of a traditional four H-bond model, as this remains the more accepted model in the literature today.

6) Page 11, lines 260-267. The discussion should take into account also the Pauli exclusion effects from comment 3 above.

We have addressed this accordingly in the 2nd paragraph on page 13.

7) Page 25, Figure 1. The comment on the blue-shift as due solely to extensive H-bond coordination should be modified in the light of the comments above. The lowest excited state in the liquid is into the antibonding 4a₁ state in XAS which is also the dominating character in optical spectroscopy of water (Brancato et al., Phys. Rev. Lett. **100**, 107401 (2008)). The blue-shift will not only relate to the number of H-bonds, but also to the strength and to the density through the excluded volume effects.

The figure caption has been changed accordingly.

Reviewer 2

This paper sets out some important outstanding questions on what makes water water, namely (p 1 - 2): "... However, debates continue over an exact definition of hydrogen bonding in water. ...", and "... Furthermore, how hydrogen bonding in a colligative sense gives rise to bulk properties is not fully understood. ...". The paper then goes on to mention sub-critical and super-critical water because of their relevance to green chemistry. Yet another thread is the connection between ultra-violet excitation of water and the production of free radicals, with "implications in radiation chemistry, atmospheric chemistry, radiation biology, and astrochemistry."

The first job of the reviewer is to ascertain whether any progress towards answering these posed questions is demonstrated in the paper.

The paper goes on to document in some detail the efforts and difficulties associated with attempting to obtain ultra-violet information about water, especially at increased temperature and pressure. Finally a new experiment is summarized using documented new methods developed at a synchrotron light source.

The results section gives an extensive account of the changes to the UV spectra with increasing temperature towards the critical point (I assume this is along the liquid-vapour equilibrium curve?) and then above the critical point at 381 C, with increasing pressure. For the former set of measurements, the spectra red-shift, with a linear decrease in extinction coefficient, suggesting a gradual reduction in the strength or degree of hydrogen bonding. For the super-critical data the spectra blue shift, and the extinction coefficient increases, in a non-linear way with increasing pressure, with some sort of transition at densities in the range 0.1 – 0.2 g/cm³. There is extensive discussion of the significant uncertainties, for example, of the extent to which water can be regarded as hydrogen bonded, associated with this region of the water phase diagram, despite many years of research. Finally attempts are made to quantify the trends seen in terms of numbers of nearest neighbour molecules as a function of density.

Yet at the end of the day, do we learn anything from the new measurements? Given all the uncertainties and mis-information about water that occurs in the literature, I would be extremely cautious about accepting on trust statements like (p 8) "It is quite possible that the extent of the red shift with increasing temperature can be correlated with weakening of the water hydrogen bond network due to added thermal energy" and (p 12) "It is highly likely that the dimers and trimers that form with increasing pressure greatly affect the electronic transition, smearing out the coherence tied to the symmetric stretch coordinate." This kind of heuristic reasoning, based on intuition rather than scientific fact, can actually be quite misleading since it is mostly not quantitative and is not supported by any kind of physical justification. Almost certainly the real system is more complicated than such simplistic statements would imply. Whereas it is relatively easy to notice and plot trends between thermodynamic variables and spectral properties, understanding what those trends actually mean in the real material is much tougher.

So I can appreciate the experimental achievement of the authors, but whether the results get us any closer to understanding the questions that are raised at the outset of the paper is not clear. The data and analysis should certainly be published, but are not appropriate for Nature Comms.

It is clear that Reviewer 2 finds the paper inappropriate for publication in *Nature Communications*, and he/she has not made any specific suggestions for improving the manuscript. Regardless, we have done our best to take his/her comments to heart in making some semantic changes to the text and support our arguments with further references especially justifying the correlation between the spectral red shift and weakening of the H-bond network.

We entirely agree with Reviewer 2 in that the real system is likely more complicated than what our modeling can address at this stage. However, we are honestly surprised to be told that our reasoning is merely heuristic in nature and without physical justification. The scientific literature is flush with studies that point out 1. the significant changes that occur to the water electronic structure upon forming water dimers and small water clusters (see references 63, 82, and 83 in particular), and 2. the weakening of the H-bond network with increasing temperature (see especially pages 8-10). In our original draft, we referenced many such articles and have now included even more for additional support. Throughout the text, we have changed the timbre of our writing to sound less suggestive and more affirmative regarding our findings, and have more explicitly addressed the data in the context of possible exciton vs. Rydbergization effects.

REVIEWERS' COMMENTS:

Reviewer #1 (Remarks to the Author):

The authors have addressed in a satisfactory way all my previous comments and I now recommend the paper for publication.

I did notice two places which the authors could reformulate:

Line 56. The potential surface is not repulsive with respect to extension of the O-H bond, but dissociative. The repulsion goes the other way, i.e. trying to shorten it if it is extended.

Lines 487-489: The last sentence is very unclear.

Reviewer 1

The authors have addressed in a satisfactory way all my previous comments and I now recommend the paper for publication.

I did notice two places which the authors could reformulate:

Line 56. The potential surface is not repulsive with respect to extension of the O-H bond, but dissociative. The repulsion goes the other way, i.e. trying to shorten it if it is extended.

Using the word “repulsive” is a clear mistake on our part. We have changed the word to “dissociative.”

Lines 487-489: The last sentence is very unclear.

We have changed the sentence in question to clarify that we are fitting a standard deviation parameter as mutual for the dimer and trimer species.